# Hydrogen spillover-driven synthesis of high-entropy alloy nanoparticles as a robust catalyst for $CO_2$ hydrogenation

Kohsuke Mori [1,2,3✉], Naoki Hashimoto[1], Naoto Kamiuchi [4], Hideto Yoshida[4], Hisayoshi Kobayashi[5] & Hiromi Yamashita [1,2,3✉]

High-entropy alloys (HEAs) have been intensively pursued as potentially advanced materials because of their exceptional properties. However, the facile fabrication of nanometer-sized HEAs over conventional catalyst supports remains challenging, and the design of rational synthetic protocols would permit the development of innovative catalysts with a wide range of potential compositions. Herein, we demonstrate that titanium dioxide ($TiO_2$) is a promising platform for the low-temperature synthesis of supported CoNiCuRuPd HEA nanoparticles (NPs) at 400 °C. This process is driven by the pronounced hydrogen spillover effect on $TiO_2$ in conjunction with coupled proton/electron transfer. The CoNiCuRuPd HEA NPs on $TiO_2$ produced in this work were found to be both active and extremely durable during the $CO_2$ hydrogenation reaction. Characterization by means of various in situ techniques and theoretical calculations elucidated that cocktail effect and sluggish diffusion originating from the synergistic effect obtained by this combination of elements.

[1] Division of Materials and Manufacturing Science, Graduate School of Engineering, Osaka University, Osaka, Japan. [2] Elements Strategy Initiative for Catalysts Batteries ESICB, Kyoto University, Kyoto, Japan. [3] Innovative Catalysis Science Division, Institute for Open and Transdisciplinary Research Initiatives (ICS-OTRI), Osaka University, Osaka, Japan. [4] The Institute of Scientific and Industrial Research, Osaka University, Osaka, Japan. [5] Kyoto Institute of Technology, Kyoto, Japan. ✉email: mori@mat.eng.osaka-u.ac.jp; yamashita@mat.eng.osaka-u.ac.jp

In contrast to conventional alloy materials based on single principal elements, high-entropy alloys (HEAs) have recently received significant attention in various research fields. These alloys represent a new class of metallic materials in which more than five near-equimolar components are mixed to form single-phase solid solutions with high mixing entropy values rather than intermetallic phases[1,2]. Various unique synergistic effects result from such mixtures, including high configuration entropy, lattice distortion, sluggish diffusion, and cocktail effects, and endow HEAs with high mechanical strength, good thermal stability, and superior corrosion resistance[3–5]. To date, several synthetic strategies have been reported, such as bulk melting[6], solid-state processing[7], and additive manufacturing[8,9], all of which have principally focused on the fabrication of bulk HEAs. However, the development of HEA nanoparticles (NPs) with a mean diameter of <10 nm lags significantly behind, despite the potential practical applications of these NPs in catalysis, nanoelectronics, and material science, owing to their large surface area-to-volume ratio and nanoscale-size effect[10].

A bottom-up approach to the fabrication of HEA NPs is likely to be more reliable than a top-down approach, because the former would be expected to produce fewer surface defects along with uniform chemical compositions and homogenous size distributions[11]. In an early study, Yao et al.[12–14] succeeded in the fabrication of HEA NPs containing up to eight elements on conductive carbon nanofibers, using a carbothermal shock method based on flash heating (at ~$10^5$ K/s) to ~2000 K followed by rapid cooling (at the same approximate rate). Subsequently, methods incorporating ultrasonication[15], solvothermal synthesis[16], polyols in solution[17,18], and fast moving bed pyrolysis[19] were explored as alternative synthetic approaches. Unfortunately, these methods still require the application of high temperatures and special experimental apparatuses. The development of new and simpler techniques for the synthesis of HEA NPs, especially those immobilized on the surfaces of conventional support materials, represents an ongoing challenge. Even so, such research could result in a wider range of industrial uses for these materials and provide a better understanding of the novel functions of nanostructured catalysts.

Hydrogen spillover is a fascinating phenomenon that occurs in sensors, hydrogen storage materials, and heterogeneous catalysis[20–22]. This process involves the surface migration of dissociated H atoms driven by a concentration gradient. Hydrogen spillover on reducible transition metal oxides such as $TiO_2$, $WO_3$, and $MoO_3$ proceeds via a set sequence of steps. These are as follows: (i) the dissociative chemisorption of $H_2$ upon interacting with a noble metal, (ii) the formation of protons ($H^+$) and electrons ($e^-$) from H atoms at metal-support interfaces, and (iii) the diffusion of these protons to lattice $O_2^-$ anions to form O–H and H–O–H bonds, accompanied by the simultaneous partial reduction of the metals in the transition metal oxide by the electrons[23,24]. Thus, in this process, H atoms migrate to adjacent hydrogen-poor metal oxide surfaces, which would not be able to dissociate $H_2$ molecules under the same conditions, via coupled proton/electron transfer. Consequently, the extent of hydrogen spillover on non-reducible supports is limited, because the simultaneous transfer of protons and electrons will not proceed on such materials[25]. Recently, van Bokhoven and colleagues[26] reported experimental data in conjunction with theoretical calculations showing that hydrogen spillover on $TiO_2$ proceeds ten orders of magnitude faster than that on the non-reducible oxide $Al_2O_3$, and that $TiO_2$ provides longer migration distances from the noble metal proton sources.

Our own group has previously demonstrated that $TiO_2$ is a promising platform for the synthesis of non-equilibrium binary alloy NPs, such as RuNi and RhCu, which are essentially immiscible at equilibrium due to the positive enthalpies of formation of their solid solution alloys[27,28]. However, the highly specific formation of binary alloy NPs based on combinations of normally immiscible noble and base metals can be achieved with the assistance of the strong spillover effect obtained from $TiO_2$. Using this oxide allows spillover hydrogen species with high reduction potentials to be generated from noble metals (Ru or Rh) and to rapidly migrate to and reduce base metals (Ni or Cu) at low temperatures. In the present work, we developed and demonstrated that this facile strategy can be applied to the synthesis of $TiO_2$-supported HEA NPs. Specifically, CoNiCuRuPd HEA NPs on $TiO_2$ displayed high activity and outstanding stability during the $CO_2$ hydrogenation reaction. This study also elucidated the specific mechanism responsible for the formation of HEA NPs, based on in situ characterization techniques. In addition, density functional theory (DFT) calculations

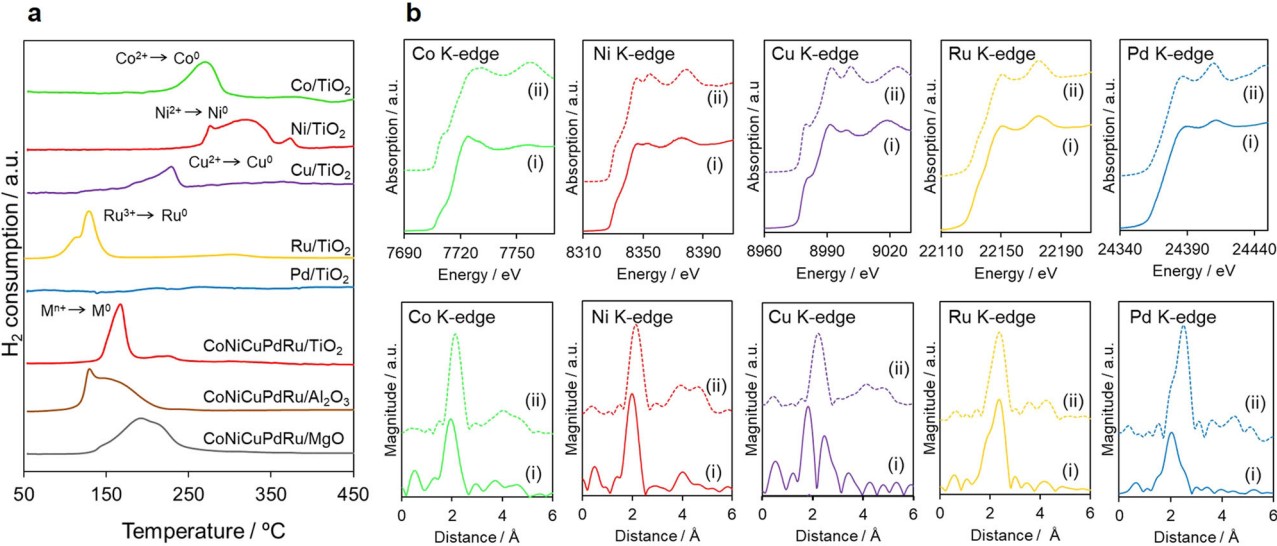

**Fig. 1 Characterization of the reduction sequence. a** $H_2$-TPR profiles for the as-deposited mono- and quinary-component samples supported on $TiO_2$, $Al_2O_3$, and MgO. **b** In situ XANES and FT-EXAFS spectra at the Co, Ni, Cu, Ru, and Pd K-edge acquired from (i) CoNiCuRuPd/$TiO_2$ after reduction under $H_2$ at 500 °C and (ii) the corresponding foil reference materials.

were performed to validate both the formation mechanism and to examine the synergistic effects of mixing multiple elements, such as unique catalytic performance and exceptional durability.

## Results

**Synthesis and characterization of HEA NPs on TiO₂.** CoNiCuRuPd HEA NPs supported on TiO₂ (CoNiCuRuPd/TiO₂) were synthesized using a simple impregnation method, employing an aqueous solution of the corresponding precursors. This was followed by reduction under a H₂ atmosphere at 400 °C without a specific calcination step before this reduction. We selected such quinary-component, because they possess different medium reduction potentials, ($E^0(Co^{2+}/Co^0) = -0.28$ V, $E^0(Ni^{2+}/Ni^0) = -0.26$ V, $E^0(Cu^{2+}/Cu^0) = +0.34$ V, $E^0(Ru^{3+}/Ru^0) = +0.46$ V, and $E^0(Pd^{2+}/Pd^0) = +0.99$ V, all vs. NHE (normal hydrogen electrode)) and Pd was used to achieve the hydrogen spillover effect. A survey of bulk multi-component alloys determined that the formation of a solid solution HEA required an atomic size difference, $\delta$, of <6.6% and an enthalpy of mixing, $\Delta H_{mix}$, between −11.6 and 3.2 kJ/mol[29]. In the present study, the CoNiCuRuPd combination met the above criteria ($\delta = 3.9$% and $\Delta H_{mix} = 1.1$ kJ/mol) and so the formation of solid solution CoNiCuRuPd HEA NPs was expected.

Figure 1a shows the H₂ temperature programmed reduction (TPR) profiles obtained from the as-deposited mono- and quinary-component samples prior to reduction under H₂. These data indicate that the single metals generated broad reduction peaks at different temperatures. In addition, the absence of a peak in the case of the Pd/TiO₂ sample suggests the immediate reduction of the deposited Pd²⁺ precursor after the switching between H₂ and Ar flows at ambient temperature[30]. The relative trend in the reduction temperatures of these materials is similar to that of the reduction potentials of the respective ions. Interestingly, the quinary-component precursors on TiO₂ generated only a single reduction peak with a maximum temperature of ~170 °C, which was an intermediate between those obtained from the mono-component samples. Thus, the reducibility of each of the Co²⁺, Ni²⁺, and Cu²⁺ ions was improved, whereas those of the Ru²⁺ and Pd²⁺ ions were decreased in comparison with the monometallic samples. This simultaneous reduction of the mixed-metal precursors indicates that all atoms were undergoing interactions with one another, leading to the formation of HEA NPs containing all five elements. In contrast, the quinary-component precursors on the non-reducible supports such as MgO and Al₂O₃ displayed broad reduction peaks ranging from 130 °C to 250 °C. These reduction profiles indicated that all atoms were not interacting on the MgO and Al₂O₃ surfaces.

In situ X-ray absorption fine structure (XAFS) analyses conducted under a H₂ atmosphere at elevated temperatures further elucidated the reduction sequence (Fig. 1b and Supplementary Figs. 1 and 2). X-ray absorption near-edge structure spectra confirmed the reduction of all the precursors at 200 °C. The intermediate shapes and edge positions at the Co and Ni K-edges indicated the presence of a mixture of cations and zero valent ions at 200 °C, due to their relatively low reduction potentials. In contrast, all spectra acquired at 400 °C resembled those of the corresponding foils, suggesting that all the elements were in a metallic state. More detailed inspection of these data also found slight changes in the post-edge region at all K-edges. As an example, the two distinct peaks at ~24,390 and 24,415 eV corresponding to the allowed $1s \rightarrow 5p$ transition at the Pd K-edge were slightly shifted to higher energy values compared to the Pd foil. This result suggested that the symmetry of the Pd metal face centered cubic (fcc) structure was slightly disordered following integration with the other metals[31].

Fourier transforms of extended XAFS (FT-EXAFS) data further clarified the structural transformation during the reduction sequence. The spectra of the as-deposited sample produced a sharp singlet peak in the K-edge region that was attributed to M–O bonds with lengths of ~1.7–1.9 Å. In the case of Co and Ni, the peak intensity due to the M–O bonds decreased at 200 °C, whereas another peak attributed to metallic M–M bonds with longer interatomic distances appeared for Cu, Ru, and Pd. These transitions demonstrated the reduction of M^n+ ions on the TiO₂ support. The bond structure after completion of the reduction revealed that the interatomic metallic M–M bond lengths were significantly different from those for the corresponding bulk references. In the case at Ru K-edge, the shouldered peak can be observed at around 1.9 Å, which is suggestive of the formation of Ru-M with shorter interatomic distances. These results suggest that all elements were surrounded by different metallic atoms.

The X-ray diffraction (XRD) pattern for CoNiCuRuPd/TiO₂ exhibited new broad peaks at $2\theta = 42.2°$ and $48.9°$. These peaks suggest the formation of a single phase with an fcc structure having a lattice parameter ($a$) of 3.734 Å, which is intermediate between 3.890 Å (for fcc Pd) and 3.524 Å (for fcc Ni) (Fig. 2a). No peaks attributable to pure Co, Ni, Cu, Ru, or Pd were detected, establishing that these components were dispersed in the NPs without segregation. Figure 2b, c present high-angle annular dark field scanning transmission electron microscopy (STEM) images

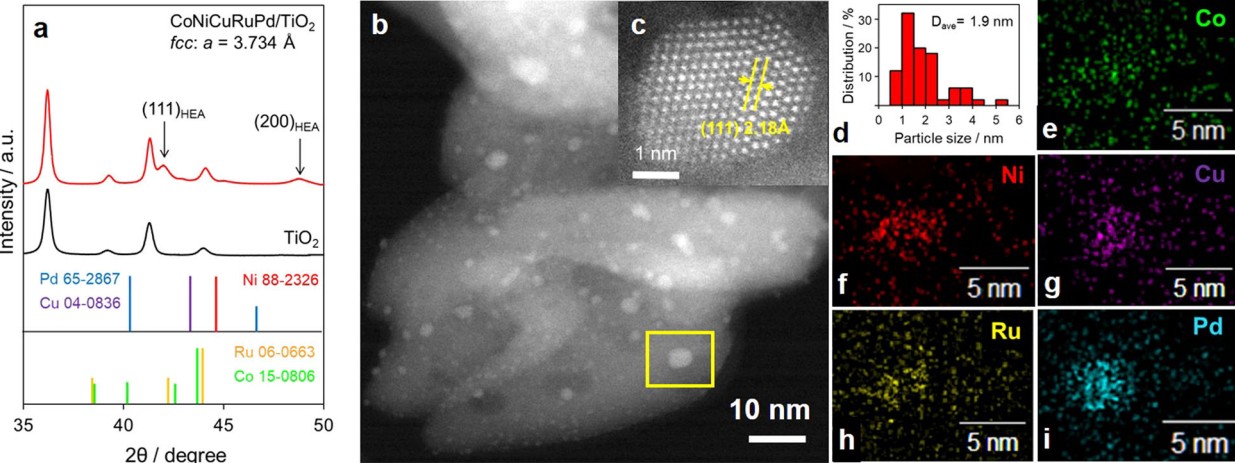

**Fig. 2 Characterization of CoNiCuRuPd/TiO₂. a** XRD pattern, **b, c** HAADF-STEM images, **d** particle size distribution, and **e–i** EDX mapping of the various elements.

showing a lattice fringe spacing of 2.18 Å. From these images, the average size ($d_{ave}$) of the CoNiCuRuPd HEA NPs was estimated to be 1.90 nm (Fig. 2d). The energy-dispersive X-ray (EDX) maps of these specimens also confirmed the distribution of each element (Fig. 2e–i). In addition, an EDX line analysis showed that all signals appeared in the same area, demonstrating the formation of a solid solution alloy involving all five elements (Supplementary Fig. 3). The $d_{ave}$ values for the CoNiCuRuPd/$Al_2O_3$ and CoNiCuRuPd/MgO samples were 6.65 and 6.73 nm, respectively (Supplementary Fig. 4), and partially segregated NPs with a bimodal particle size distribution were observed on the MgO support. These results suggest that the $TiO_2$ support ensured more rapid and homogeneous reduction at lower temperatures, allowing the formation of nuclei to provide smaller, uniform HEA NPs without segregation.

**Formation mechanism driven by hydrogen spillover over $TiO_2$.** Considering the TPR with $H_2$ ($H_2$-TPR) and in situ XAFS results, we propose a mechanism for the formation of the HEA NPs on the $TiO_2$ support in conjunction with hydrogen spillover (Fig. 3a). In this process, under a $H_2$ atmosphere, the $Pd^{2+}$ precursors are first partially reduced to generate nuclei. Following this, $H_2$ is dissociated on the surfaces of these Pd nuclei to form Pd−H species (Step 1). The reduction of $Ti^{4+}$ to $Ti^{3+}$ together with the transfer of H atoms from Pd nuclei at the metal-support interfaces (Step 2) is accompanied by the migration of electrons from $Ti^{3+}$ ions to neighboring $Ti^{4+}$ ions. This promotes the subsequent simultaneous transfer of protons to $O^{2-}$ anions attached to these adjacent $Ti^{4+}$ ions (Step 3). In this manner, the hydrogen atoms rapidly reach all metal ions by moving over the $TiO_2$ surface (Step 4), such that these ions are all reduced at the same time to form the HEA NPs (Step 5), accompanied with the regeneration of $Ti^{4+}$. In situ XRD pattern of CoNiCuRuPd/$TiO_2$ after reduction under $H_2$ at 200 °C showed the broad peak due to the Pd (111) at 40.5° and Pd (200) at 47.0°, respectively (Supplementary Fig. 5), which disappeared with increasing the reduction temperature, accompanied with the appearance of new peaks due to the formation of HEA NPs (Fig. 2a). In situ FT-EXAFS spectra at the Pd K-edge of CoNiCuRuPd/$TiO_2$ acquired under $H_2$ at 200 °C showed the strong peak due to contagious Pd–Pd bond, which slightly shifted toward shorter interatomic distance with increasing the temperature (Supplementary Fig. 2e). These are clear evidence for the formation of Pd nuclei in the early stage, which act as uptake sites to enhance the migration of active hydrogen atoms. These results also indicated the initial formation of $Pd_{core}$−$M_{shell}$ (M represents Co, Ni, Cu, or Ru metal) structure, which finally forms HEA NPs via atomic diffusion with increasing the reduction temperature, owing to the increase of configuration entropy.

This mechanism based on the spillover effect was further evaluated by DFT calculations, using rutile $TiO_2$ (101) as a model because of its thermodynamic stability and $Pd_5$ clusters as a model for Pd nuclei. According to the above proposed reaction mechanism, four representative elementary steps were considered for the reduction of metal cations on the $TiO_2$ through the hydrogen spillover from Pd clusters. The resulting potential energy profile is shown in Fig. 3b. Moving along this profile, the dissociation of $H_2$ on a $Pd_5$ cluster (Step 1) occurs with a barrier of 20.1 kcal/mol. The activation energy ($E_a$) associated with subsequent H atom transfer from a $Pd_5$ cluster to a neighboring O atom on the support (Step 2) was estimated to be 27.5 kcal/mol. Owing to the presence of oxygen sites having different coordination numbers, such as 2-coordinated oxygen (O(2)) or 3-coordinated oxygen (O(3)) sites, the migration of a H atom over the $TiO_2$ surface (Step 3) was calculated separately for each

scenario. The activation energies for the migration of a H atom from O(2) to O(2), O(2) to O(3), and O(3) to O(2) sites were determined to be 15.0, 37.4, and 12.7 kcal/mol, respectively (Supplementary Fig. 6). These data demonstrate that the participation of O(3) sites in the migration of H atoms over the $TiO_2$(101) is energetically unfavorable and so this migration preferentially occurs at O(2) sites because of the abundance of such sites on $TiO_2$(101). The reduction of other deposited metal cations by the spilled H atoms (Step 4) was further evaluated by calculating $E_a$ for the attack of a neighboring H atom on an $M^{n+}$−OH species (M = $Co^{2+}$, $Ni^{2+}$, $Cu^{2+}$, $Ru^{3+}$, or $Pd^{2+}$) on the support, together with the loss of $H_2O$. These $E_a$ values were estimated to be 12.7, 12.7, 12.6, 8.0, and 7.4 kcal/mol for $Co^{2+}$, $Ni^{2+}$, $Cu^{2+}$, $Ru^{3+}$, and $Pd^{2+}$, respectively. Thus, this step had the lowest energy requirement for all cations in the overall reaction. The energy barrier in Step 5 was determined to be <11.1 kcal/mol by considering the formation energy of a five-nucler cluster model containing five different elements from each reduced atom (Supplementary Fig. 7). This is lower than those in Steps 1–3, suggesting that the reduced atoms are easily migrating to form HEA NPs.

The dissociation energy of a gaseous $H_2$ molecule on $TiO_2$(101) without Pd clusters was estimated to be 82.3 kcal/mol (Supplementary Fig. 8). It was also calculated that the direct reduction of $Co^{2+}$ on $TiO_2$(101) by a gaseous $H_2$ molecule occurred with a barrier of 85.3 kcal/mol (Supplementary Fig. 9), which was more than six times greater than that for the same process with a spilled H atom. This preliminary analysis further confirmed that spilled H atoms in the presence of Pd clusters promoted the rapid and simultaneous reduction of the multiple metal precursors at low temperatures on a thermodynamic basis. In comparison, Step 3 on hexagonal $Al_2O_3$(100) was found to be thermodynamically unfavorable, with $E_a$ values of 29.3, 41.7, and 43.8 kcal/mol for the transfer pathways from O(2) to O(3), O(3) to O(2), and O(3) to O(3) sites, respectively (Supplementary Fig. 10), which were more than twice as great as those for $TiO_2$(101). Similar calculations were also performed using the γ-$Al_2O_3$ model proposed by Digne et al.[32], who reported an $E_a$ for hydrogen migration (38.9 kcal/mol) that was similar to our result for hexagonal $Al_2O_3$[33]. These results clearly suggest that H atom transfer on $TiO_2$ was energetically more likely to proceed than that on $Al_2O_3$; hence, the rate of hydrogen spillover was faster on the $TiO_2$.

**Catalytic $CO_2$ hydrogenation.** The hydrogenation of $CO_2$ to high calorific fuels has the potential to alleviate both climate change and future demands for fossil fuels[34,35]. As an example, the endothermic reverse water–gas shift reaction ($CO_2 + H_2 \rightarrow CO + H_2O$, $\Delta H = 41$ kJ/mol) is one of the most promising means of producing CO as an important feedstock for Fischer–Tropsch processes and as an intermediary step for the further synthesis of fuel and chemicals[36,37]. In addition, the exothermic $CO_2$ methanation reaction ($CO_2 + 4H_2 \rightarrow CH_4 + 2H_2O$, $\Delta H = -165.0$ kJ/mol), also known as the Sabatier reaction, has attracted new interest because of the recent development of the power-to-gas concept[38,39]. This reaction is also recognized as an important approach to powering long-term space exploration missions[40].

In the present work, catalytic performance was evaluated based on monitoring the progress of atmospheric pressure $CO_2$ hydrogenation at temperatures from 300 to 400 °C, with CO and $CH_4$ as the major products (Fig. 4a). CoNiCuRuPd/$TiO_2$ gave the highest yield of hydrogenated products, which was from 2 and 13 times greater, respectively, than those obtained using MgO and $Al_2O_3$ as supports. This enhanced activity can presumably be ascribed to the formation of a quinary-component HEA NPs solely on the $TiO_2$, as indicated by the $H_2$-TPR data. Specifically,

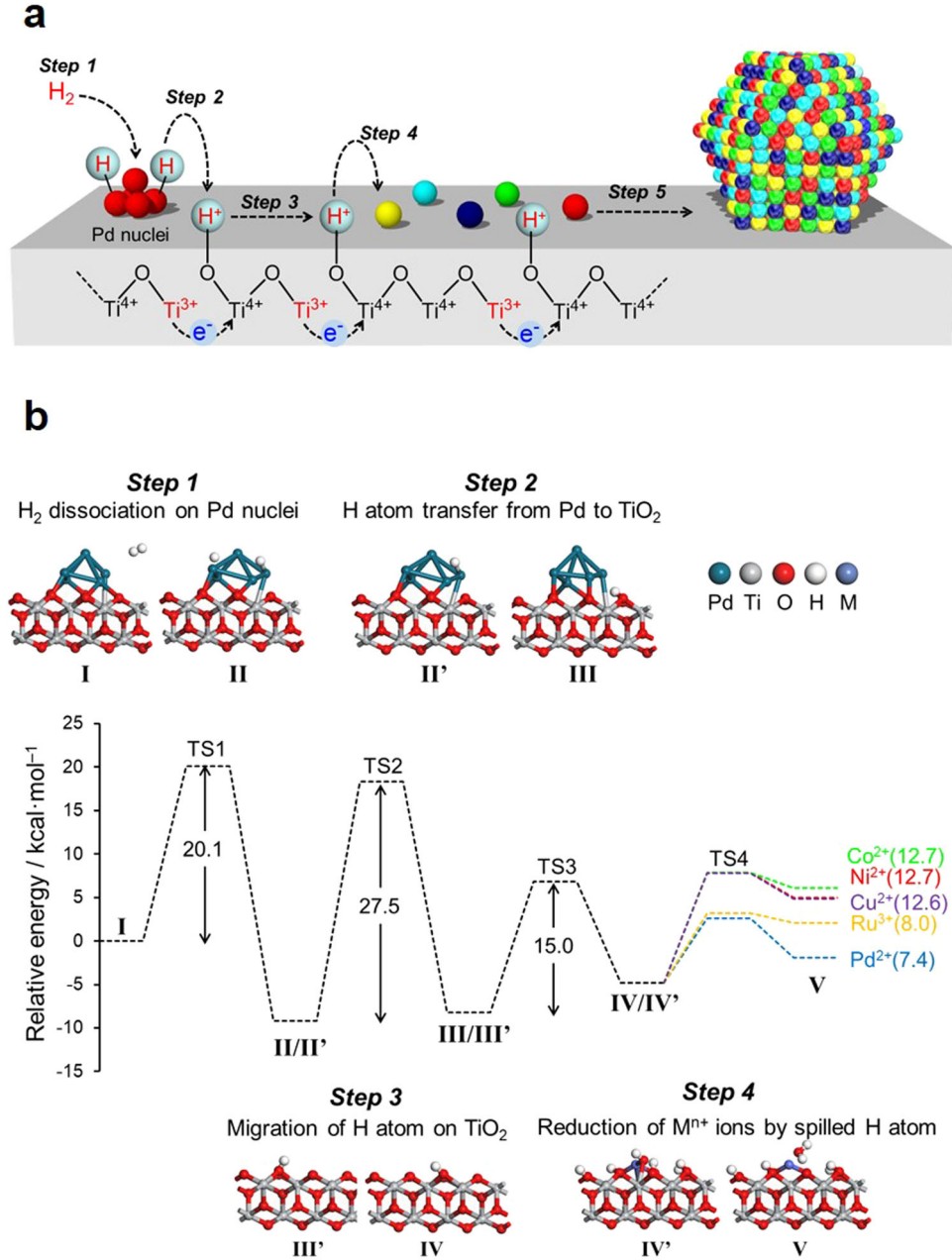

**Fig. 3 Formation mechanism of HEA NPs on a TiO$_2$ support assisted by hydrogen spillover. a** Schematic illustration of the elementary steps and **b** the potential energy profile of processes on the TiO$_2$(101) as obtained from DFT calculations. The values in parentheses are the calculated energy barriers for each M$^{n+}$ cation in Step 4.

the quinary-component precursors on the MgO and Al$_2$O$_3$ displayed broad reduction peaks ranging from 130 °C to 250 °C, suggesting the formation of larger segregated NPs rather than smaller HEA NPs. The selectivities for CO and CH$_4$ were also found to vary depending on the catalyst that was employed. CoNiCuRuPd/TiO$_2$ showed relatively high selectivity for CH$_4$ (68.3% CH$_4$ selectivity at 400 °C), similar to that of the Al$_2$O$_3$ specimen (72.1% CH$_4$ selectivity at 400 °C) but quite different from that obtained using MgO (75.2% CO selectivity at 400 °C). It should also be noted that the catalytic activity of Pd/TiO$_2$ prepared by the same method was low compared with that of CoNiCuRuPd/TiO$_2$, and that this monometallic sample gave CO as the primary product. As shown in Fig. 4b, an apparent activation energy ($E_a$) of 37.7 kJ/mol was obtained for

CoNiCuRuPd/TiO$_2$, which was lower than that of 44.2 kJ/mol for Pd/TiO$_2$. As shown in Supplementary Fig. 11a, the reaction using Co/TiO$_2$, Ni/TiO$_2$, and Cu/TiO$_2$ hardly occurred, in which the yields of hydrogenated products were <1%. The monometallic Ru/TiO$_2$ showed relatively high activity, in which yields of hydrogenated products were 48.5%, 61.6%, and 66.1% for 300 °C, 350 °C, and 400 °C, respectively, with the predominant formation of CH$_4$. However, these high activity is presumably ascribed to the extremely small Ru NPs ($d_{ave}$ = 0.89 nm) (Supplementary Fig. 11b). These results clearly suggest the so-called cocktail effect originating from the synergistic effect obtained from the combination of elements comprising the HEA.

At atmospheric pressure, the most widely accepted mechanism for CO$_2$ hydrogenation is initiated by the adsorption and

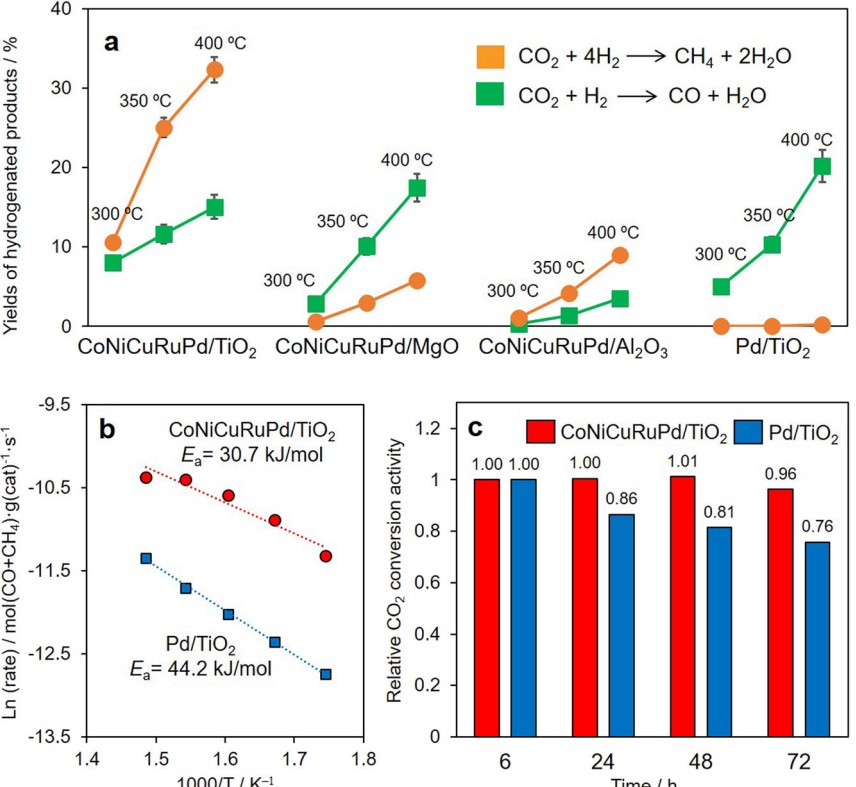

**Fig. 4 Comparison of catalytic activities during atmospheric pressure $CO_2$ hydrogenation. a** Yields of hydrogenated products over quinary-component alloys supported on $TiO_2$, MgO, and $Al_2O_3$, and over $Pd/TiO_2$ at various reaction temperatures, **b** Arrhenius plots obtained from $CO_2$ hydrogenation data over CoNiCuRuPd/$TiO_2$ and Pd/$TiO_2$, and **c** relative activities over time, showing the durability of CoNiCuRuPd/$TiO_2$ and Pd/$TiO_2$ during $CO_2$ hydrogenation.

activation of $CO_2$ at the metal/oxide interfaces of the metal-supported catalyst[41,42]. Hydrogenation and/or dissociation subsequently occur to afford a chemically adsorbed CO intermediate that is either desorbed as a product or undergoes further hydrogenation to form $CH_4$. Previous studies have demonstrated that both catalytic activity and selectivity are affected by the particle size of the active metal centers and by the metal/support interfaces[43]. As the particles sizes in CoNiCuRuPd/$TiO_2$ ($d_{ave} = 1.90$ nm) and Pd/$TiO_2$ ($d_{ave} = 2.04$ nm) were similar (Supplementary Fig. 12), the different selectivities for CO or $CH_4$ observed in this study were primarily attributed to the desorption characteristics of CO molecules at metal sites with different binding strengths.

For this reason, the surfaces of the NPs were assessed using temperature-programed desorption (TPD) with adsorbed CO, together with Fourier transform infrared spectroscopy. In the case of Pd/$TiO_2$, a peak assignable to the linear stretching vibration of adsorbed CO ($v_{CO}$) was observed at 2091 cm$^{-1}$ in association with the initiation of CO desorption at 50 °C (Fig. 5a). In contrast, this $v_{CO}$ peak was observed at 2070 cm$^{-1}$ in the spectrum obtained from CoNiCuRuPd/$TiO_2$. This shift toward a lower wavenumber occurred together with a change in the CO desorption temperature to above 150 °C (Fig. 5b). These results readily explain the selectivity observed during $CO_2$ hydrogenation over these materials. The adsorption sites on CoNiCuRuPd/$TiO_2$ were definitely electron enriched compared with those on the monometallic Pd/$TiO_2$. This, in turn, delayed the desorption of the CO intermediate owing to the stronger interactions, thus promoting subsequent hydrogenation to form $CH_4$[42]. These experimental results were also supported by theoretical DFT

calculations. The frequency of CO adsorbed on fcc CoNiCuRuPd was modeled using randomly populated (111) facets of periodically repeating slab models (with the 15 configuration patterns depicted in Supplementary Fig. 13)[44], giving an average $v_{CO}$ of 2079 cm$^{-1}$. The adsorption energies ($E_{ad}$) of CO and H on an fcc surface, fcc hollow and hexagonal close packed (hcp) hollow were also calculated for CoNiCuRuPd(111) and for pure metal slabs (Supplementary Fig. 14). The average $E_a$ values for CO and H adsorption on CoNiCuRuPd HEA (denoted as HEA$_{ave}$ (111) in Fig. 5c) were determined to be −37.5 and −50.3 kcal/mol, respectively. The average $E_a$ for CO adsorption on Pd(111) was substantially lower at −26.2 kcal/mol, whereas the $E_a$ for H (−54.2 kcal/mol) was similar. These results demonstrate that the interaction between CO and the HEA surface was stronger than that with the Pd surface, suggesting that $CH_4$ and CO would be preferentially formed on the former and latter, respectively. It should be further noted that the average $E_a$ for CO and H adsorption on all the pure metals (denoted as Ave$_{CoNiCuRuPd}$ in Fig. 5c) was different from the HEA$_{ave}$ (111). This result provided additional evidence for a cocktail effect originating from the synergistic effect of the combined metals, which gives rise to unique electronic properties.

**Structural robustness of HEA NPs.** Another crucial phenomenon associated with HEA NPs that affects catalytic performance is the sluggish diffusion effect, which enhances the durability of the catalyst. In trials with Pd/$TiO_2$, the catalytic activity during $CO_2$ hydrogenation was found to gradually decrease with continued use, such that the relative activity was reduced by a factor of 0.76 after a 72 h reaction (Fig. 4c). Similarly, the activity of the

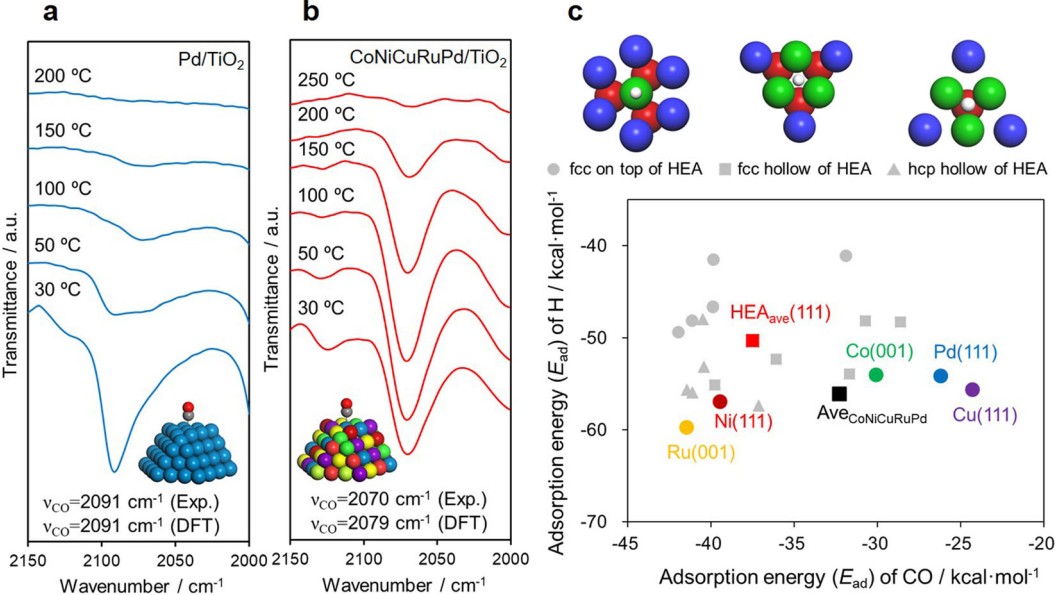

**Fig. 5 Comparison of adsorption characteristics.** FTIR data obtained during the TPD of adsorbed CO on (**a**) $Pd/TiO_2$ and (**b**) $CoNiCuRuPd/TiO_2$. In these calculations, a correction coefficient of 1.074 was applied to adjust the vibrational frequency of CO adsorbed on the surface of an fcc Pd(111) slab model to the experimental value of 2091 cm$^{-1}$[158]. **c** $E_{ad}$ values calculated for CO and H on an upper fcc surface (●), an fcc hollow (■), and an hcp hollow (▲) for CoNiCuRuPd(111) and for pure metal slabs (Co(001), Cu(111), Ni(111), Ru(001), and Pd(111)). Here, HEA$_{ave}$ (111) and Ave$_{CoNiCuRuPd}$ are the averages of the $E_a$ values for CO and H adsorption on the CoNiCuRuPd HEA and on the pure metal slabs.

Ru/TiO$_2$ decreased by a factor of 0.89 after a 72 h reaction (Supplementary Fig. 11c). In contrast, CoNiCuRuPd/TiO$_2$ retained 96% of its original activity, while keeping a constant selectivity. Each of these catalyst specimens was recovered after 72 h and subjected to a TEM analysis (Supplementary Fig. 15). A substantial enlargement of the NPs was observed in the case of Pd/TiO$_2$, such that the average NP diameter was more than doubled to 5.3 nm from 2.0 nm. Conversely, CoNiCuRuPd/TiO$_2$ exhibited suppressed particle growth and the mean particle diameter was determined to be 2.3 nm (Supplementary Fig. 16). The homogenous elemental distribution evident in the EDX mapping data also provided strong evidence for the maintenance of the random HEA structure. In addition, EDX line scans confirmed that single NPs contained all the constituent elements.

The structural robustness of the HEA NPs was also confirmed by monitoring radiation damage process using TEM under electron beam irradiation in vacuum[45,46]. Here, the contrast of atomic positions was analyzed in the continuous image. As shown in the time-lapsed TEM images, the change of the contrast in the atomic column position is relatively small for the CoNiCuRuPd/TiO$_2$ even at edge/corner position (Fig. 6a–c), indicating the suppression of structure deterioration by an incident electron beam. Conversely, drastic changes in contrast were observed for Pd/TiO$_2$, which is definitely originated from the atomic displacement induced by the knock-on damage (Fig. 6d–f)[47,48]. The temporal changes in intensity of atomic columns at other positions showed similar trend, as summarized in Supplementary Fig. 17. The statistic and precise analysis is indispensable for discussing the number of atoms at an atomic column from the contrast of a TEM image[49,50]. Nevertheless, the stability of the surface atoms in the CoNiCuRuPd NPs has a clear difference from the monometallic Pd NPs.

In an effort to better understand the high robustness of the HEA NPs, theoretical investigations were conducted employing cluster models. DFT calculations demonstrated that the cohesive energy ($E_c$) of a Co$_{16}$Ni$_{15}$Cu$_{16}$Ru$_{16}$Pd$_{16}$ HEA cluster was −3.92 eV, which was higher than the value of −3.09 eV for a Pd$_{79}$ cluster (Fig. 7a). Combining these data with molecular dynamics (MD) simulations, diffusion coefficients ($D$) were determined at 900 K after 0.1 ps (Supplementary Fig. 18). The results demonstrated that the $D$ values of all metals in a Co$_{16}$Ni$_{15}$Cu$_{16}$Ru$_{16}$Pd$_{16}$ HEA cluster were lower than those for the corresponding monometallic clusters (Co$_{79}$, Cu$_{79}$, Ni$_{79}$, Ru$_{79}$, or Pd$_{79}$) (Fig. 7b). As an example, the $D$ for Pd in a Co$_{16}$Ni$_{16}$Cu$_{15}$Ru$_{16}$Pd$_{16}$ HEA cluster was calculated to be $1.31 \times 10^{-5}$ m$^2$/s, and so was approximately one-third lower than the value of $3.43 \times 10^{-5}$ m$^2$/s for a Pd$_{79}$ cluster. These results provide further evidence that sluggish diffusion in the HEA NPs, originating from the mixing of multiple elements and from lattice distortion effects, contributed significantly to the high resistance of the HEA NPs against the undesired irreversible agglomeration and radiation damage process.

## Discussion

We succeeded in the facile low-temperature synthesis of supported HEA NPs, taking advantage of the hydrogen spillover that proceeds on TiO$_2$ via a coupled proton–electron transfer mechanism. Both in situ observations and theoretical simulations provided evidence that Pd$^{2+}$ ions are first reduced by H$_2$ to generate nuclei, after which the dissociation of hydrogen molecules occurs to form active hydrogen atoms that enable the simultaneous reduction of neighboring precursors. A CoNiCuRuPd/TiO$_2$ catalyst synthesized in this manner exhibited different selectivity and significantly improved stability compared with Pd/TiO$_2$ during the hydrogenation of CO$_2$. Theoretical investigations also emphasized that the sluggish diffusion in these CoNiCuRuPd HEA NPs is caused by the combination of multiple metals, and that lattice distortion plays a crucial role in the superior robustness of this material. The preliminary H$_2$-TPR studies of the quinary-component precursors including Rh, Pt, and Au on TiO$_2$ generated only a single reduction peak (Supplementary Fig. 19), which is suggestive of the formation of HEA

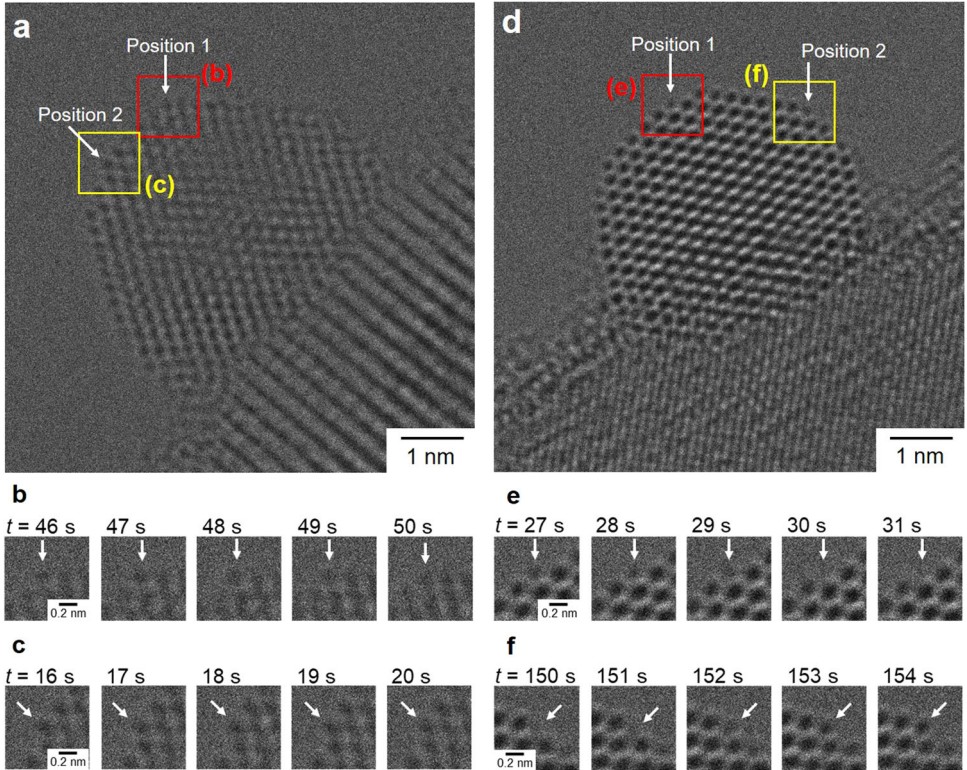

**Fig. 6 Stability of surface atoms under electron beam irradiation.** A representative TEM image of a nanoparticle in **a** HEA/TiO$_2$ and **d** Pd/TiO$_2$, and **b**, **c**, **e**, **f** sequential images of surface atoms taken from TEM movies (Supplementary Movies 1 and 2).

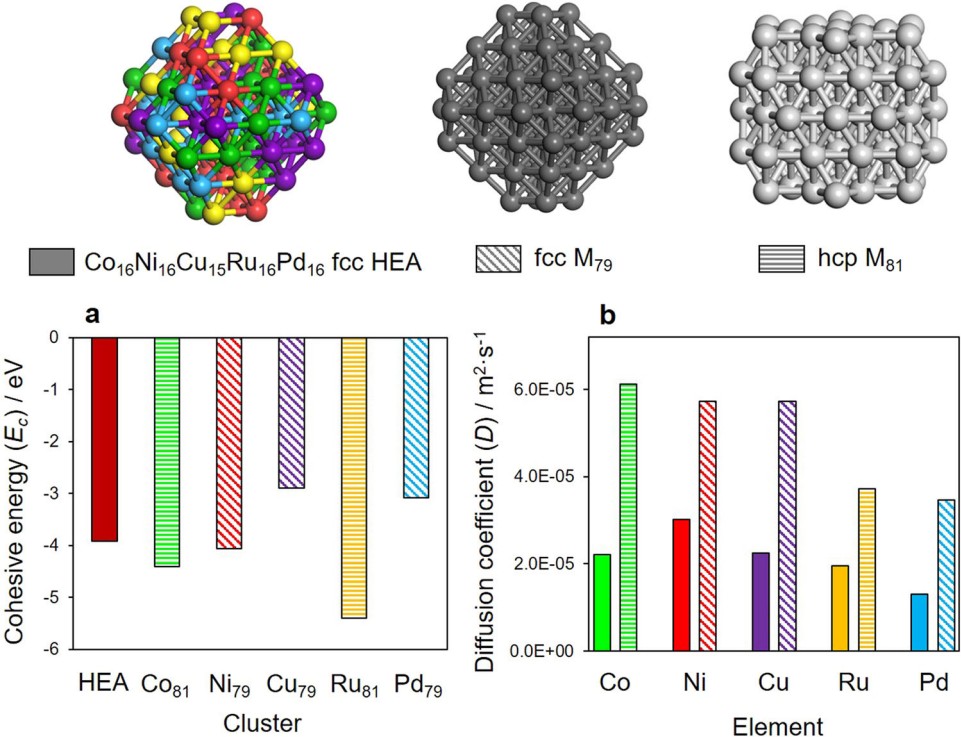

**Fig. 7 Sluggish diffusion in HEA NPs. a** Cohesive energies of Co$_{16}$Ni$_{16}$Cu$_{15}$Ru$_{16}$Pd$_{16}$ HEA and M$_{79}$ model clusters as calculated using DFT and **b** atomic diffusion coefficients for the elements in the Co$_{16}$Ni$_{16}$Cu$_{15}$Ru$_{16}$Pd$_{16}$ HEA and M$_{79}$ models as determined by MD simulations.

NPs by the simultaneous reduction of the mixed-metal precursors. Further investigation to demonstrate the applicability of the present synthetic method is now under investigation. This study demonstrates not only an ideal heterogeneous catalyst based on HEA NPs with durability that suggests potential practical applications but also offers advanced insights into an innovative catalyst/photocatalyst architecture providing an essentially unlimited compositional space.

## Methods

**Materials**. Rutile TiO₂ (JRC-TIO-6) was supplied by the Catalysis Society of Japan. Al₂O₃ was obtained from Stream Chemicals, Inc., whereas MgO was purchased from Wako Pure Chemical Industries, Ltd. RuCl₃·$n$H₂O, Cu(NO₃)₂·3H₂O, Co (NO₃)₂·6H₂O, and Ni(NO₃)₂·6H₂O were purchased from Nacalai Tesque, and Na₂PdCl₄ was obtained from the Tokyo Chemical Industry Co., Ltd. All commercially available compounds were used as received.

**Preparation of catalysts**. TiO₂ (0.5 g) was dispersed in distilled water (100 mL) followed by the addition of 10 mM of each metal precursor solution (4.75 mL). This mixture was stirred at room temperature for 1 h, after which the water was evaporated under vacuum. Finally, the sample was reduced under a 20 mL/min flow of H₂ at 400 °C for 2 h to yield CoNiCuRuPd/TiO₂ (Pd 1.0 wt%; Co:Ni:Cu:Ru:Pd = 1:1:1:1:1 [on a molar basis]). CoNiCuRuPd/Al₂O₃ (Pd 1.0 wt%; Co:Ni:Cu:Ru:Pd = 1:1:1:1:1), CoNiCuRuPd/MgO (Pd 1.0 wt%; Co:Ni:Cu:Ru:Pd = 1:1:1:1:1), and Pd/TiO₂ (Pd 1.0 wt.%) with the same metal loadings were also synthesized according to the same procedure.

**Characterization**. TEM micrographs were obtained with a field-emission TEM instrument (Hf-2000, Hitachi) equipped with an EDX detector (Kevex) operated at 200 kV. STEM images, elemental mapping, and line analysis were obtained using a JEOL-ARM 200F instrument equipped with a Kevex EDX detector (JED 2300T) operated at 200 kV. H₂-TPR was conducted using a BEL-CAT (BEL Japan, Inc.) instrument by heating 50 mg samples at 5 °C/min from 50 °C to 600 °C under a 5.0% H₂/Ar flow. These analyses were performed using as-deposited samples before H₂ reduction. A TPD study using adsorbed CO was performed with a JASCO FT/IR-6600 instrument. In addition, in situ XAFS spectra and XRD patterns were acquired at the 01B1 beamline station in conjunction with a Si (111) monochromator at SPring-8, JASRI, Harima, Japan (proposal numbers 2019A1048 and 2019B1091). In a typical experiment, spectra were acquired, whereas a pellet sample was held in a batch-type in situ XAFS cell. XAFS data were processed using the REX2000 software program (Rigaku).

**Computational method**. Adsorption energies, $E_{ad}$, were calculated using the DFT, employing DMol³ program[51,52] with Materials Studio 17.2 interface. The generalized gradient approximation exchange-correlation functional proposed by Perdew, Burke and Ernzerhof (PBE)[53] was combined with the double numerical plus polarization basis sets. A slab consisting of a 4 × 4 surface unit cell was adopted. The slab consists of three atomic (111) layers. The geometry of bottom two layers was fixed at the corresponding bulk positions, and that of top layer and adsorbate was allowed to relax during geometry optimizations. The lattice constant to the surface normal direction was taken to 30 Å including the vacuum region. $E_{ad}$ was defined by the equation $E_{ad} = E_{adsorbate/slab} - (E_{adsorbate} + E_{slab})$, where $E_{adsorbate/slab}$, $E_{adsorbate}$, and $E_{slab}$ are the total energies of adsorbate–slab system, free adsorbate, and bare slab, respectively.

Simulations for the formation mechanism of HEA NPs via H₂ spillover were performed using a TiO₂(101) slab with 2 × 2 surface unit cell and three-layer thickness was constructed with a vacuum thickness of 20 Å, on which a square pyramidal Pd₅ cluster was loaded. The top-layer atoms were allowed to relax during geometry optimizations and the other layers were fixed at the corresponding bulk positions.

For the cohesive energy ($E_c$) calculation of pure metal and HEA clusters composed of Co, Ni, Cu, Ru, and Pd, the plane wave-based program Castep was employed[54,55]. The PBE functional was used together with the ultrasoft-core potentials[56]. The basis set cutoff energy was set to 351 eV. The electron configurations of the atoms were Co: 3d⁷ 4s², Ni: 3d⁸ 4s², Cu: 3d¹⁰ 4s¹, Ru: 4s² 4p⁶ 4d⁷ 5s¹, and Pd: 4d¹⁰. Sphere-like 79 and 81 atom clusters were used for FCC and HCP metals, and the clusters were placed in a cubic cell with a side of 30 Å. For the alloy cluster preparation, Pd atoms in Pd₇₉ cluster were randomly replaced by Co, Ni, Cu, and Ru atoms, and Co₁₆Ni₁₅Cu₁₆Ru₁₆Pd₁₆ cluster was built. $E_c$ was defined by the equation $E_c = (E_{cluster} - mE_{atom})/m$, where $E_{cluster}$ and $E_{atom}$ are the total energies of the pure metal or alloy cluster, and isolated single atom, respectively. $m$ is the total number of atoms.

DFT-based MD calculations were also performed to estimate the difference in diffusion coefficients ($D$) between pure metal and HEA clusters employing Castep. First, the structures of pure metal and HEA clusters were optimized, and then the optimized structures were subjected to MD calculations. The conditions are

microcanonical (NVE) ensemble, 900 K, time step: 1 fs, and 100 steps. $D$ was evaluated from the mean-square displacement according to Eq. (1)

$$D = \frac{1}{6} \frac{\left\langle |r(t_2) - r(t_1)|^2 \right\rangle}{t_2 - t_1} \tag{1}$$

where $t_1$ and $t_2$ are the initial and final times of simulation interval. The $D$ was evaluated between the start and end ($t_1 = 0$ and $t_2 = 100$) of simulation. The intermediate $D$ values per each ten steps were also calculated to check the convergence of simulation.

**Catalytic activity trials**. The performance of each catalyst was evaluated using a fixed-bed reactor system in which a portion of catalyst (50 mg) was placed into a quartz cell with an internal diameter of 17 mm, held within an electric oven. The as-prepared catalyst was pretreated by heating at 5 °C/min to 400 °C in a flow of H₂ (20 mL/min) for 2 h. The sample was subsequently exposed to a N₂/H₂/CO₂ mixture having a 4/5/1 composition (total flow of 50 mL/min, SV = 6000 mL/g/h). Reaction products were analyzed online using a gas chromatograph (Shimadzu GC-14B) equipped with an active carbon column connected to a thermal conductivity detector followed by a flame ionization detector equipped with a methanizer.

**Environmental TEM (ETEM) observation**. The effect of electron irradiation was monitored using an ETEM apparatus (Titan ETEM G2, Thermo Fisher Scientific, Inc., USA) with a Cs-corrector of the objective lens, a monochromator, and a K3-IS Direct Detection camera (Gatan, Inc., USA). The accelerating voltage and electron current flux were set at 300 kV and 2 A/cm², respectively. The observation in this condition does not cause serious damage to TiO₂ support[57]. The base pressure around specimen was kept below 1 × 10⁻⁵ Pa.

## Data availability

All data generated and analyzed during this study are included in this article and its Supplementary Information, or are available from the corresponding authors upon reasonable request.

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

## Acknowledgements
The present work was supported by the Grant-in-Aid for Scientific Research from the Ministry of Education, Culture, Sports, Science and Technology (MEXT) of Japan (numbers T19K220820 and A18H020740), Element Strategy Initiative of MEXT, Japan (number JPMXP0112101003), and "Dynamic Alliance for Open Innovation Bridging Human, Environment and Materials" from MEXT.

## Author contributions
K.M. supervised the project and wrote the manuscript. N.H. performed the catalyst preparation and characterization. K.M. and H.K. performed MD calculations. N.K. and H.Y. performed the ETEM experiment. H.Y. helped supervise the project. The manuscript was written through the discussion with all authors. All authors have given approval to the final version of manuscript.

## Competing interests
The authors declare no competing interests.
