## [Peer Review File · Nature Communications]

REVIEWER COMMENTS

Reviewer #1 (Remarks to the Author):

In this work, the authors successfully obtained CoNiCuRuPd high entropy alloy (HEA) nanoparticles (NPs) based on the hydrogen spillover over TiO₂. The formation mechanism via hydrogen spillover effect was further elucidated by density functional theory (DFT) calculations. Besides, the solid-solution CoNiCuRuPd NPs exhibited impressive catalytic performance towards CO₂ hydrogenation. The mechanism of the unique catalytic activity, selectivity, and stability of CoNiCuRuPd NPs were also systematically investigated by DFT calculations, time-lapsed TEM, molecular dynamics (MD) simulations, and so on. This work is suggested to be published in Nat. Commun. after addressing the following issues. Specific comments are as follows:

1. In CoNiCuRuPd HEA NPs, Pd was used to achieve the hydrogen spillover effect. Why the elements Co, Ni, Cu, and Ru were selected to fabricate this HEA?
2. The in situ EXAFS spectra for Co, Ni, Cu, and Ru elements are obviously distinct from that of corresponding foil reference materials, while the main peak of the in situ EXAFS spectra for Ru K-edge seems to be similar to that of Ru foil. A Table of fitted coordination number and bond length for the elements are suggested to be provided to better elucidate the lattice distortion in HEA.
3. The formation process of CoNiCuRuPd needs to be further elucidated. Based on the basic steps of hydrogen spillover effect, the reduction of Mn⁺ was indeed induced by the electrons accumulated during the diffusion of protons in transition metal oxide, which was not reflected from Figure 3A. Besides, whether the generated Ti³⁺ would be oxidized to Ti⁴⁺ after the reduction of Mn⁺?
4. As the authors mentioned, Pd²⁺ was firstly reduced to form Pd nuclei. In this process, Pd clusters or core-shell structure with Pd core may be formed. XPS spectra of CoNiCuRuPd/TiO₂ needs to be provided.
5. The catalytic performance of CoNiCuRuPd/TiO₂ was compared with that of Pd/TiO₂ to illustrate the cocktail effect of HEA. How about the catalytic performance of Co/TiO₂, Ni/TiO₂, Cu/TiO₂, or Ru/TiO₂? The catalytic performance of CoNiCuRuPd/TiO₂ should at least be compared with that of these catalysts to further validate the potential synergistic effect of HEA.
6. There are some grammatical errors in the manuscript. The main text should be carefully checked.

Reviewer #2 (Remarks to the Author):

In the manuscript, the authors reported their experimental and computational results of the fabrication

of high-entropy alloy (HEA) CoNiCuRuPd nanoparticles and the electrochemical activity and stability of these alloy nanoparticle catalysts for CO₂ hydrogenation reaction. Specifically, the authors have synthesized the HEA nanoparticles by reducing the metal precursors in an aqueous solution and under a H₂ atmosphere, characterized the structure of the HEA nanoparticles using XRD, STEM, and TEM techniques, measured the activity and durability of the catalysts for CO₂ hydrogenation reaction, and carried out the density functional theory (DFT) calculations to predict the energy evolution of the formation of HEA nanoparticles. Based on their results, the authors concluded that they have developed a novel approach for HEA nanoparticle synthesis and the thus-synthesized HEA nanoparticles exhibit enhanced stability during electrochemical reactions. The topic of the manuscript is of current interests, however the main conclusions derived in the manuscript are not sufficiently justified. Consequently, this reviewer does not believe the current manuscript should be considered for publication in the Nature Communications.

My major criticisms to the manuscript are given in below.

1. One main conclusion of the manuscript is that the authors have developed a new synthesis method for fabrication of HEA nanoparticles. However, the authors failed to demonstrate the wide application of the synthesis approach since only one alloy system was reported.
2. The manuscript lacks its focus. As suggested by the title and abstract, the manuscript should focus on understanding of the HEA synthesis approach. However, the manuscript spends a significant portion to report the observed catalytic performance of the HEA nanoparticles. The authors are suggested to re-organize their presentation to make the focus clear.
3. The characterization results presented in Figure 2 are not sufficient to show the random distribution of different metal element in a single particle.
4. The presented DFT results in Figure 3 only show the pathway of the hydrogen dissociation and subsequent hydrogen migration on TiO₂. In order to form a sizable HEA nanoparticle, the neutral metal atoms are also required to migrate on TiO₂. The energy evolution for this process is missing in the current study.

Reviewer #3 (Remarks to the Author):

The manuscript describes the use of hydrogen spillover to generate high entropy alloys. This is an interesting manuscript that nicely shows how to make nano-sized multi-metal particles on a reducible support. Palladium is used as a source to dissociate hydrogen after Pd²⁺ reduction; via spillover, other metal ions are reduced simultaneously generating the HEA. The HEA are more stable than Pd only particles in the conversion of carbon dioxide.

There are a few issues that need to be addressed before I can suggest to accept the manuscript:
The authors describe multiple steps that must be fulfilled, Figure 3B. Two things remain unclear: 1. Experimental evidence of the structure of the palladium nuclei must be provided; what is the structure? And 2. How are the HAE formed from the ion precursors, that are dispersed. In other words, how do the ions/atoms migrate and end up in a single particle? DFT may provide support for a model.

Figure 6 and corresponding movies are beautiful, however, given the role of beam-induced effects may be not relevant for the manuscript, which claims chemical stability. I suggest to not add such data to the current manuscript.

A final minor comment:

Only one author name (ref 26) mentioned; either everywhere or nowhere.

We would like to thank all the referees for the careful review and the valuable comments, which allowed us to improve the paper. Below we list the changes we have made in light of the referees' comments.

Answers to the comments by Referee 1

Over all comment: In this work, the authors successfully obtained CoNiCuRuPd high entropy alloy (HEA) nanoparticles (NPs) based on the hydrogen spillover over TiO₂. The formation mechanism via hydrogen spillover effect was further elucidated by density functional theory (DFT) calculations. Besides, the solid-solution CoNiCuRuPd NPs exhibited impressive catalytic performance towards CO₂ hydrogenation. The mechanism of the unique catalytic activity, selectivity, and stability of CoNiCuRuPd NPs were also systematically investigated by DFT calculations, time-lapsed TEM, molecular dynamics (MD) simulations, and so on. This work is suggested to be published in Nat. Commun. after addressing the following issues. Specific comments are as follows:

Answer: We would like to thank referee 1 for the careful review and the positive comments.

Comment 1: In CoNiCuRuPd HEA NPs, Pd was used to achieve the hydrogen spillover effect. Why the elements Co, Ni, Cu, and Ru were selected to fabricate this HEA?

Answer 1: In order to demonstrate the advantage of the hydrogen spillover-driven low temperature synthesis of HEA NPs, we selected Co, Ni, Cu, and Ru because they possess different medium reduction potentials, ($E^0(\text{Co}^{2+}/\text{Co}^0) = -0.28$ V, $E^0(\text{Ni}^{2+}/\text{Ni}^0) = -0.26$ V, $E^0(\text{Cu}^{2+}/\text{Cu}^0) = +0.34$ V, $E^0(\text{Ru}^{3+}/\text{Ru}^0) = +0.46$ V and $E^0(\text{Pd}^{2+}/\text{Pd}^0) = +0.99$ V, all vs. NHE). The higher reduction temperatures may be required for the use of other cations with more negative reduction potentials, such as Mn ($E^0(\text{Mn}^{2+}/\text{Mn}^0) = -1.05$ V), Fe ($E^0(\text{Fe}^{2+}/\text{Fe}^0) = -0.44$ V), and Zn ($E^0(\text{Zn}^{2+}/\text{Zn}^0) = -0.76$ V). Such experiments are now under investigation in our laboratory.

In order to clarify the above reason, the following sentences were added.

(Page 3) CoNiCuRuPd HEA NPs supported on TiO₂ (CoNiCuRuPd/TiO₂) were synthesized using a simple impregnation method, employing an aqueous solution of the corresponding precursors. This was followed by reduction under a H₂ atmosphere at 400 °C without a specific calcination step before this reduction. We selected such quinary-component because they possess different medium reduction potentials, ($E^0(\text{Co}^{2+}/\text{Co}^0) = -0.28$ V, $E^0(\text{Ni}^{2+}/\text{Ni}^0) = -0.26$ V, $E^0(\text{Cu}^{2+}/\text{Cu}^0) = +0.34$ V, $E^0(\text{Ru}^{3+}/\text{Ru}^0) = +0.46$ V and $E^0(\text{Pd}^{2+}/\text{Pd}^0) = +0.99$ V, all vs. NHE) and Pd was used to achieve the hydrogen spillover effect.

Comment 2: The in situ EXAFS spectra for Co, Ni, Cu, and Ru elements are obviously distinct from that of corresponding foil reference materials, while the main peak of the in situ EXAFS spectra for Ru K-edge seems to be similar to that of Ru foil. A Table of fitted coordination number and bond length for the elements are suggested to be provided to better elucidate the lattice distortion in HEA.

Answer 2: The curve fitting analysis of the inverse FT-EXAFS spectra is useful to investigate the local structure of the alloy NPs. However, it is required a wide variety of shell parameter to define the curve fitting analysis for the HEA NPs with five different ionic radii., which complicate the understanding of the obtained results. Thus, we think that such investigation does not always provide better elucidation in lattice distortion of HEA.

The *in situ* FT-EXAFS spectra of CoNiCuRuPd/TiO₂ at Ru K-edge shows the shouldered peak at around 1.9 Å due to the formation of Ru-M with shorter interatomic distances, which cannot be observed in the Ru foil.

In order to clarify the above discussions, the following sentences were modified as follows.

(Page 4) The bond structure after completion of the reduction revealed that the interatomic metallic M–M bond lengths were significantly different from those for the corresponding bulk references. In the case at Ru K-edge, the shouldered peak can be observed at around 1.9 Å, which is suggestive of the formation of Ru-M with shorter interatomic distances. These results suggest that all elements were surrounded by different metallic atoms.

Comment 3: The formation process of CoNiCuRuPd needs to be further elucidated. Based on the basic steps of hydrogen spillover effect, the reduction of Mⁿ⁺ was indeed induced by the electrons accumulated during the diffusion of protons in transition metal oxide, which was not reflected from Figure 3A. Besides, whether the generated Ti³⁺ would be oxide to Ti⁴⁺ after the reduction of Mⁿ⁺?

Answer 3: Thank you very much for reviewer's comment. According to the comment, the electron was added in **Figure 3A**. As referee pointed out, the generated Ti³⁺ would be oxide to Ti⁴⁺ after the reduction of Mⁿ⁺. In order to clarify the above issue, **Figure 3A** and the following sentence was modified.

(Page 5) In this manner, the hydrogen atoms rapidly reach all metal ions by moving over the TiO₂ surface (**Step 4**), such that these ions are all reduced at the same time to form the HEA NPs (**Step 5**), accompanied with the regeneration of Ti⁴⁺.

Comment 4: As the authors mentioned, Pd²⁺ was firstly reduced to form Pd nuclei. In this process, Pd clusters or core-shell structure with Pd core may be formed. XPS spectra of CoNiCuRuPd/TiO₂ needs to be provided.

Answer 4: Thank you very much for reviewer's comment. *In situ* XRD pattern of CoNiCuRuPd/TiO₂ after reduction under H₂ at 200 °C showed the broad peak due to the Pd (111) at 40.5 ° and Pd (200) at 47.0 °, respectively (**Figure S5**), which disappeared with increasing the reduction temperature, accompanied with the appearance of new peaks due to the formation of HEA NPs (**Figure 2A**). These are clear evidence for the formation of Pd nuclei at the initial stage, which are randomly distributed during the migration of other reduced atoms to form HEA NPs.

In order to clarify the above result and discussions, **Figure S5** and the following sentences were added.

Figure S5. In situ XRD pattern of CoNiCuRuPd/TiO₂ after reduction under H₂ at 200 °C.

(Page 5) *In situ* XRD pattern of CoNiCuRuPd/TiO₂ after reduction under H₂ at 200 °C showed the broad peak due to the Pd (111) at 40.5 ° and Pd (200) at 47.0 °, respectively (**Figure S5**), which disappeared with increasing the reduction temperature, accompanied with the appearance of new peaks due to the formation of HEA NPs (**Figure 2A**). These are clear evidence for the formation of Pd nuclei in the early stage, which act as uptake sites to enhance the migration of active hydrogen atoms.

Comment 5: The catalytic performance of CoNiCuRuPd/TiO₂ was compared with that of Pd/TiO₂ to illustrate the cocktail effect of HEA. How about the catalytic performance of Co/TiO₂, Ni/TiO₂, Cu/TiO₂, or Ru/TiO₂? The catalytic performance of CoNiCuRuPd/TiO₂ should at least be compared with that of these catalysts to further valid the potential synergistic effect of HEA.

Answer 5: According to the reviewer's comment, the catalytic activity of the monometallic catalysts prepared by the same method was performed. As shown in **Figure S11A**, the reaction using Co/TiO₂, Ni/TiO₂, and Cu/TiO₂ hardly occurred, in which the yields of hydrogenated products were less than 1%. On the other hand, the monometallic Ru/TiO₂ showed relatively high activity, in which yields of hydrogenated products were 48.5 %, 61.6 %, and 66.1% for 300 °C, 350 °C, and 400 °C, respectively with the predominant formation of CH₄. These high activity is presumably ascribed to the extremely small Ru NPs. TEM analysis of the Ru/TiO₂ showed that the average diameter (d_{ave}) is 0.89 nm (**Figure S11B**), which is significantly smaller than that of the CoNiCuRuPd/TiO₂ (d_{ave} = 1.90 nm). This does not allow the fair comparison. Therefore, we compared the catalytic activity, surface property, and durability of the CoNiCuRuPd/TiO₂ with the monometallic Pd/TiO₂ because of its similar particle size (d_{ave} = 2.04 nm).

In order to clarify these results and discussions, the **Figure 11X** and the following sentences were added.

(Page 7) As shown in **Figure S11A**, the reaction using Co/TiO₂, Ni/TiO₂, and Cu/TiO₂ hardly occurred, in which the yields of hydrogenated products were less than 1%. The monometallic Ru/TiO₂ showed relatively high activity, in which yields of hydrogenated products were 48.5 %, 61.6 %, and 66.1% for 300 °C, 350 °C, and 400 °C, respectively with the predominant formation of CH₄. But these high activity is presumably ascribed to the extremely small Ru NPs (d_{ave} = 0.89 nm) (**Figure S11B**).

Figure S11. (A) Yields of hydrogenated products over monometallic Co, Ni, Cu, and Ru-supported TiO₂, (B) TEM image and size distribution diagrams of Ru/TiO₂.

Comment 6: There are some grammatical errors in the manuscript. The main text should be carefully checked.

Answer 6: This manuscript was carefully checked and revised by native speaker.

Answers to the comments by Referee 2

Over all comment: In the manuscript, the authors reported their experimental and computational results of the fabrication of high-entropy alloy (HEA) CoNiCuRuPd nanoparticles and the electrochemical activity and stability of these alloy nanoparticle catalysts for CO₂ hydrogenation reaction. Specifically, the authors have synthesized the HEA nanoparticles by reducing the metal precursors in an aqueous solution and under a H₂ atmosphere, characterized the structure of the HEA nanoparticles using XRD, STEM, and TEM techniques, measured the activity and durability of the catalysts for CO₂ hydrogenation reaction, and carried out the density functional theory (DFT) calculations to predict the energy evolution of the formation of HEA nanoparticles. Based on their results, the authors concluded that they have developed a novel approach for HEA nanoparticle synthesis and the thus-synthesized HEA nanoparticles exhibit enhanced stability during electrochemical reactions. The topic of the manuscript is of current interests, however the main conclusions derived in the manuscript are not sufficiently justified. Consequently, this reviewer does not believe the current manuscript should be considered for publication in the Nature Communications. My major criticisms to the manuscript are given in below.

Answer: We would like to thank referee 2 for the careful review and the positive comments.

Comment 1: One main conclusion of the manuscript is that the authors have developed a new synthesis method for fabrication of HEA nanoparticles. However, the authors failed to demonstrate the wide application of the synthesis approach since only one alloy system was reported.

Answer 1: Thank you very much for the reviewer's useful comment. In this study, we employed Co, Ni, Cu, Ru, and Pd as components. Pd ($E^0(\text{Pd}^{2+}/\text{Pd}^0) = +0.99$ V, all vs. NHE) was used to achieve the hydrogen spillover effect, and we selected Co, Ni, Cu, and Ru because they possess different medium reduction potentials, ($E^0(\text{Co}^{2+}/\text{Co}^0) = -0.28$ V, $E^0(\text{Ni}^{2+}/\text{Ni}^0) = -0.26$ V, $E^0(\text{Cu}^{2+}/\text{Cu}^0) = +0.34$ V, and $E^0(\text{Ru}^{3+}/\text{Ru}^0) = +0.46$ V). The higher reduction temperatures may be required for the use of other cations with more negative reduction potentials, such as Mn ($E^0(\text{Mn}^{2+}/\text{Mn}^0) = -1.05$ V), Fe ($E^0(\text{Fe}^{2+}/\text{Fe}^0) = -0.44$ V), and Zn ($E^0(\text{Zn}^{2+}/\text{Zn}^0) = -0.76$ V). Such experiments are now under investigation in our laboratory.

In order to clarify the above reason, the following sentences were added.

(Page 3) CoNiCuRuPd HEA NPs supported on TiO₂ (CoNiCuRuPd/TiO₂) were synthesized using a simple impregnation method, employing an aqueous solution of the corresponding precursors. This was followed by reduction under a H₂ atmosphere at 400 °C without a specific calcination

step before this reduction. We selected such quinary-component because they possess different medium reduction potentials, ($E^0(\text{Co}^{2+}/\text{Co}^0) = -0.28$ V, $E^0(\text{Ni}^{2+}/\text{Ni}^0) = -0.26$ V, $E^0(\text{Cu}^{2+}/\text{Cu}^0) = +0.34$ V, $E^0(\text{Ru}^{3+}/\text{Ru}^0) = +0.46$ V and $E^0(\text{Pd}^{2+}/\text{Pd}^0) = +0.99$ V, all vs. NHE) and Pd was used to achieve the hydrogen spillover effect.

Moreover, the preliminary investigation was performed to demonstrate the applicability of the present synthetic method using Rh, Pt, and Au as a component instead of Ru. Such combination meet the required criteria the formation of a solid solution HEA (size difference, $\delta < 6.6\%$ and an enthalpy of mixing, -11.6 kJ/mol $< \Delta H_{\text{mix}} < 3.2$ kJ/mol). As shown in **Figure S19**, the H₂-TPR profile of the quinary-component precursors including Rh, Pt, and Au on TiO₂ generated only a single reduction peak, which is suggestive of the formation of HEA NPs by the simultaneous reduction of the mixed-metal precursors. Further investigation is now under investigation in our laboratory.

In order to clarify these results and discussions, the **Figure S19** and the following sentences were added in the discussion section.

Figure S19. H₂-TPR characterization of the reduction sequence for the as-deposited quinary-component samples supported on TiO₂. Such combination meet the required criteria the formation of a solid solution HEA (size difference, $\delta < 6.6\%$ and an enthalpy of mixing, -11.6 kJ/mol $< \Delta H_{\text{mix}} < 3.2$ kJ/mol).

(Page 9) The preliminary H₂-TPR studies of the quinary-component precursors including Rh, Pt, and Au on TiO₂ generated only a single reduction peak (**Figure S19**), which is suggestive of the formation of HEA NPs by the simultaneous reduction of the mixed-metal precursors. Further investigation to demonstrate the applicability of the present synthetic method is now under investigation.

Comment 2: The manuscript lacks its focus. As suggested by the title and abstract, the manuscript should focus on understanding of the HEA synthesis approach. However, the manuscript spends a significant portion to report the observed catalytic performance of the HEA nanoparticles. The authors are suggested to re-organize their presentation to make the focus clear.

Answer 2: Thank you for the reviewer's useful comment. We re-considered the title and abstract in order to reflect the content of our manuscript as follows.

Manuscript title

Old: Hydrogen Spillover-Driven Low Temperature Synthesis of High-Entropy Alloy Nanoparticles as a Robust Catalyst

New: Hydrogen Spillover-Driven Synthesis of High-Entropy Alloy Nanoparticles as a Robust Catalyst for CO₂ hydrogenation

Abstract

High-entropy alloys (HEAs) have been intensively pursued as potentially advanced materials because of their exceptional properties. However, the facile fabrication of nanometer-sized HEAs over conventional catalyst supports remains challenging, and the design of rational synthetic protocols would permit the development of innovative catalysts with a wide range of potential compositions. Herein, we demonstrate that titanium dioxide (TiO₂) is a promising platform for the low-temperature synthesis of supported CoNiCuRuPd HEA nanoparticles (NPs) at 400 °C. This process is driven by the pronounced hydrogen spillover effect on TiO₂ in conjunction with coupled proton/electron transfer. ~~In this process, Pd nuclei generated in the early stage act as uptake sites to enhance the migration of active hydrogen atoms, and the five component metals are simultaneously reduced by spilled hydrogen on the support rather than via direct reduction by gaseous H₂.~~ The CoNiCuRuPd HEA NPs on TiO₂ produced in this work were found to be both active and extremely durable during the CO₂ hydrogenation reaction. ~~Characterization by means of various *in situ* techniques and theoretical calculations elucidated that cocktail effect and sluggish diffusion originating from the synergistic effect obtained by this combination of elements.~~

Comment 3: The characterization results presented in Figure 2 are not sufficient to show the random distribution of different metal element in a single particle.

Answer 3: We suppose that the formation of HEA NPs is supported by not only the EDX mapping (**Figure 2(E)-(I)**), but also the EDX line analysis along the single NPs (**Figure S3**). Because of the small NPs size ($d_{ave} = 1.9$ nm), the signals are relatively low. But all signals appeared in the same area, which demonstrates the formation of a solid solution alloy involving all five elements. In order to demonstrate the inclusion of five elements in a single NP, EDX mapping analysis of other area was also added in **Figure S3**.

In addition, we characterised the obtained CoNiCuRuPd HEA NPs on TiO₂ by XRD (**Figure 2A**), HAADF-STEM (**Figure 2C**), and *in situ* XAFS (**Figure 1B**) analysis. Judging from all data comprehensively, we can conclude the formation of HEA NPs involving all five elements. But our data do not demonstrate the completely homogeneous HEA NPs, as referee pointed out. Therefore, we modified **Figure S3** and some sentences in the manuscript as follows.

(Page 4)

Old: No peaks attributable to pure Co, Ni, Cu, Ru or Pd were detected, establishing that these components were ~~homogeneously~~ dispersed in the NPs without segregation.

New: No peaks attributable to pure Co, Ni, Cu, Ru or Pd were detected, establishing that these components were dispersed in the NPs without segregation.

(Page 5)

Old: The energy dispersive X-ray (EDX) maps of these specimens also confirmed the ~~homogeneous~~ distribution of each element (**Figures 2E-I**).

New: The energy dispersive X-ray (EDX) maps of these specimens also confirmed the distribution of each element (**Figures 2E-I**).

Figure S3. Characterization of the CoNiCuRuPd/TiO₂. (A) HAADF-STEM image of CoNiCuRuPd/TiO₂, (B) HAADF-STEM image of Area 1, (C)-(G) EDX mapping of the corresponding elements, (H) EDX line analysis along the arrow in (B). (H) HAADF-STEM image of Area 2, (I)-(M) EDX mapping of the corresponding elements

Comment 4: The presented DFT results in Figure 3 only show the pathway of the hydrogen dissociation and subsequent hydrogen migration on TiO₂. In order to form a sizable HEA nanoparticle, the neutral metal atoms are also required to migrate on TiO₂. The energy evolution for this process is missing in the current study.

Answer 4: The energy barrier in the formation of the HEA NPs from the reduced each atom (**Step 5**) was calculated by DFT, where Co-Ni-Cu-Ru-Pd 5-nucler cluster was employed as a model of HEA NP. As shown in **Figure S7**, the activation energy was determined to be less than 13.1 kcal/mol. This is lower than those in **Step 1-3**, suggesting that the reduced atoms are easily migrating to form HEA NP.

In order to clarify the above results, **Figure S7** was added. Additionally, the following sentences were modified.

(Page 5) In this manner, the hydrogen atoms rapidly reach all metal ions by moving over the TiO₂ surface (**Step 4**), such that these ions are all reduced at the same time to form the HEA NPs (**Step 5**), accompanied with the regeneration of Ti⁴⁺.

(Page 6) The energy barrier in **Step 5** was determined to be less than 11.1 kcal/mol by considering the formation energy of 5-nucler cluster model containing 5 different elements from the reduced each atom (**Figure S7**). This is lower than those in **Step 1-3**, suggesting that the reduced atoms are easily migrating to form HEA NPs.

Figure S7. The potential energy profile of **Step 5** (formation energy of 5-nucler cluster model containing 5 different elements from the reduced each atom) on the TiO₂ (101) obtained by DFT calculations. The values are calculated energy barriers.

Answers to the comments by Referee 3

Over all comment: The manuscript describes the use of hydrogen spillover to generate high entropy alloys. This is an interesting manuscript that nicely shows how to make nano-sized multi-metal particles on a reducible support. Palladium is used as a source to dissociate hydrogen after Pd²⁺ reduction; via spillover, other metal ions are reduced simultaneously generating the HEA. The HEA are more stable than Pd only particles in the conversion of carbon dioxide. There are a few issues that need to be addressed before I can suggest to accept the manuscript:

Answer: We would like to thank referee 2 for the careful review and the positive comments.

Comment 1: The authors describe multiple steps that must be fulfilled, Figure 3B. Two things remain unclear: 1. Experimental evidence of the structure of the palladium nuclei must be provided; what is the structure? And 2. How are the HAE formed from the ion precursors, that are dispersed. In other words, how do the ions/atoms migrate and end up in a single particle? DFT may provide support for a model.

Answer 1-1: Thank you very much for reviewer's comment. *In situ* XRD pattern of CoNiCuRuPd/TiO₂ after reduction under H₂ at 200 °C showed the broad peak due to the Pd (111) at 40.5 ° and Pd (200) at 47.0 °, respectively (**Figure S5**), which disappeared with increasing the reduction temperature, accompanied with the appearance of new peaks due to the formation of HEA NPs (**Figure 2A**). These are clear evidence for the formation of Pd nuclei at the initial stage, which are randomly distributed during the migration of other reduced atoms to form HEA NPs.

In order to clarify the above result and discussions, **Figure S5** and the following sentences were added.

Figure S5. In situ XRD pattern of CoNiCuRuPd/TiO₂ after reduction under H₂ at 200 °C.

(Page 5) *In situ* XRD pattern of CoNiCuRuPd/TiO₂ after reduction under H₂ at 200 °C showed the broad peak due to the Pd (111) at 40.5 ° and Pd (200) at 47.0 °, respectively (**Figure S5**), which

disappeared with increasing the reduction temperature, accompanied with the appearance of new peaks due to the formation of HEA NPs (**Figure 2A**). These are clear evidence for the formation of Pd nuclei in the early stage, which act as uptake sites to enhance the migration of active hydrogen atoms.

Answer 1-2: Please see the answer 4 for reviewer 2.

Comment 2: Figure 6 and corresponding movies are beautiful, however, given the role of beam-induced effects may be not relevant for the manuscript, which claims chemical stability. I suggest to not add such data to the current manuscript.

Answer 2: In this study, the TEM measurement under electron beam irradiation was performed to confirm the structural robustness of the HEA NPs. As referee has pointed out, such chemical stability does not directly relevant to the activity and stability under catalytic reactions, but we believe that this experiment emphasizes the specific robustness of the HEA NPs for the first time and that our findings are of sufficiently immediate interest to a general scientific research readership. Therefore, we leave these data in the manuscript as they are.

Comment 3: Only one author name (ref 26) mentioned; either everywhere or nowhere.

Answer 3: ref 26 was provided all authors.

26. Karim, W., Spreafico, C., Kleibert, A., Gobrecht, J., VandeVondele, J., Ekinci, Y. & van Bokhoven, J. A. Catalyst Support Effects on Hydrogen Spillover. *Nature* **541**, 68-71 (2017).

REVIEWERS' COMMENTS

Reviewer #1 (Remarks to the Author):

Overall, the authors responded to our comments convincingly and this work has been improved. But I still care about the EXAFS results and X-ray photoelectron spectroscopy (XPS) measurements of CoNiCuRuPd/TiO₂. This work can be published in Nat. Commun. after addressing the following issues:

1. As the author claimed, Ru/TiO₂ catalysts showed superior catalytic performance than that of CoNiCuRuPd/TiO₂, which may be ascribed to the difference of particle size. Nevertheless, during the synthesis of CoNiCuRuPd/TiO₂, small Ru clusters may also form on the surface of CoNiCuRuPd alloy, which may be the active centers for CO₂ hydrogenation. The curve fitting analysis of EXAFS in this work may complicated the issue of disorder of HEA, but this problem should be better elucidated. The reference (Sci. Adv. 2020, 6, eaaz0510) may be useful for the authors.

2. The in situ XRD results indeed proved the formation of Pd nuclei. Thus, it is possible that Pd-M (M represents Co, Ni, Cu, or Ru metal) core-shell structure may be formed during the synthetic process. The XPS spectra of CoNiCuRuPd/TiO₂ still needs to be provided.

Reviewer #2 (Remarks to the Author):

The quality of the manuscript has been greatly improved. All my previous concerns have been properly addressed. This reviewer recommends the manuscript to be accepted for publication as its current form.

We would like to thank all the referees for the careful review and the valuable comments, which allowed us to improve the paper. Below we list the changes we have made in light of the referees' comments.

Answers to the comments by Referee 1

Over all comment: The authors responded to our comments convincingly and this work has been improved. But I still care about the EXAFS results and X-ray photoelectron spectroscopy (XPS) measurements of CoNiCuRuPd/TiO₂. This work can be published in Nat. Commun. after addressing the following issues.

Answer: We would like to thank referee 1 for the careful review and the positive comments.

Comment 1: As the author claimed, Ru/TiO₂ catalysts showed superior catalytic performance than that of CoNiCuRuPd/TiO₂, which may be ascribed to the difference of particle size. Nevertheless, during the synthesis of CoNiCuRuPd/TiO₂, small Ru clusters may also form on the surface of CoNiCuRuPd alloy, which may be the active centers for CO₂ hydrogenation. The curve fitting analysis of EXAFS in this work may complicated the issue of disorder of HEA, but this problem should be better elucidated. The reference (Sci. Adv. 2020, 6, eaaz0510) may be useful for the authors.

Answer 1: Thank you for the reviewer's careful comment. As pointed out, curve fitting analysis of EXAFS may useful to elucidate the local structure of NPs. But, we can exclude the formation of small Ru cluster on the surface of CoNiCuRuPd alloy as an active center for CO₂ hydrogenation by the following experimental results.

Firstly, the monometallic Ru/TiO₂ predominantly formed CH₄ (**Figure S11A**), while The selectivities for CO and CH₄ were 31.7% and 68.3% for CoNiCuRuPd/TiO₂ at 400 °C.

Secondary, the catalytic activity of the monometallic Ru/TiO₂ was found to gradually decrease with continued use, and the relative activity was reduced by a factor of 0.89 after a 72 h reaction (**Figure S11C**). In contrast, CoNiCuRuPd/TiO₂ retained 96% of its original activity.

If small Ru clusters were formed on the surface of CoNiCuRuPd alloy, the activity and durability of Ru/TiO₂ and CoNiCuRuPd/TiO₂ should be similar.

Moreover, the shouldered peak can be observed at around 1.9 Å in the case at Ru K-edge FT-EXAFS (**Figure 1B**), which is suggestive of the formation of Ru-M with shorter interatomic distances.

In order to clarify the above discussion, **Figure S11C** and the following sentences were added.

Figure S11. (A) Yields of hydrogenated products over monometallic Co, Ni, Cu, and Ru-supported TiO₂, (B) TEM image and size distribution diagrams of Ru/TiO₂, and (C) relative activities over time, showing the durability of CoNiCuRuPd/TiO₂ and Ru/TiO₂ during CO₂ hydrogenation.

(Page 8) In trials with Pd/TiO₂, the catalytic activity during CO₂ hydrogenation was found to gradually decrease with continued use, such that the relative activity was reduced by a factor of 0.76 after a 72 h reaction (Figure 4C). Similarly, the activity of the Ru/TiO₂ decreased by a factor of 0.89 after a 72 h reaction (Figure S11C). In contrast, CoNiCuRuPd/TiO₂ retained 96% of its original activity, while keeping constant selectivity. Each of these catalyst specimens was recovered after 72 h and subjected to a TEM analysis (Figure S15).

Comment 2: The in situ XRD results indeed proved the formation of Pd nuclei. Thus, it is possible that Pd-M (M represents Co, Ni, Cu, or Ru metal) core-shell structure may be formed during the synthetic process. The XPS spectra of CoNiCuRuPd/TiO₂ still needs to be provided.

Answer 2: The XPS spectra provide the useful information about the oxidation state of the surface metals, but we do not have the in situ equipment to investigate the surface characteristic during the synthetic process. On the other hand, in situ XRD and XAFS analysis were performed under H₂ atmosphere at elevated temperature, which provides more useful information in the present study.

In situ XRD pattern of CoNiCuRuPd/TiO₂ after reduction under H₂ at 200 °C showed the broad peak due to the Pd (111) at 40.5 ° and Pd (200) at 47.0 °, respectively (**Figure S5**), which disappeared with increasing the reduction temperature, accompanied with the appearance of new peaks due to the formation of HEA NPs (**Figure 2A**). *In situ* FT-EXAFS spectrum at the Pd K-edge of CoNiCuRuPd/TiO₂ acquired under H₂ at 200 °C showed the strong peak due to contiguous Pd-Pd bond, which slightly shifted toward shorter interatomic distance with increasing the temperature (**Figure S2(e)**). These are clear evidence for the formation of Pd nuclei in the early stage, which act as uptake sites to enhance the migration of active hydrogen atoms. As referee pointed out, our results also indicated the initial formation of Pd_{core}-M_{shell} (M represents Co, Ni, Cu, or Ru metal) structure, which finally transforms into HEA NPs via atomic diffusion with increasing the reduction temperature owing to the increase of configuration entropy.

In order to clarify the above discussions, the following sentences were modified.

(Page 5) *In situ* XRD pattern of CoNiCuRuPd/TiO₂ after reduction under H₂ at 200 °C showed the broad peak due to the Pd (111) at 40.5 ° and Pd (200) at 47.0 °, respectively (**Figure S5**), which disappeared with increasing the reduction temperature, accompanied with the appearance of new peaks due to the formation of HEA NPs (**Figure 2A**). *In situ* FT-EXAFS spectra at the Pd K-edge of CoNiCuRuPd/TiO₂ acquired under H₂ at 200 °C showed the strong peak due to contiguous Pd-Pd bond, which slightly shifted toward shorter interatomic distance with increasing the temperature (**Figure S2(e)**). These are clear evidence for the formation of Pd nuclei in the early stage, which act as uptake sites to enhance the migration of active hydrogen atoms. These results also indicated the initial formation of Pd_{core}-M_{shell} (M represents Co, Ni, Cu, or Ru metal) structure, which finally forms HEA NPs via atomic diffusion with increasing the reduction temperature owing to the increase of configuration entropy.